# The association between power outages and cardiovascular and respiratory hospitalizations among US Medicare beneficiaries in 2018: A case-crossover study

Heather McBrien[1], Daniel Mork[2], Vivian Do[1], Marianthi-Anna Kioumourtzoglou[1], Joan A. Casey[3,4]*

1 Department of Environmental Health Sciences, Columbia Mailman School of Public Health, New York, New York, United States of America, 2 Department of Biostatistics, Harvard T. H. Chan School of Public Health, Boston, Massachusetts, United States of America, 3 University of Washington Department of Environmental and Occupational Health Sciences, Seattle, Washington, United States of America, 4 University of Washington Department of Epidemiology, Seattle, Washington, United States of America

* jacasey@uw.edu

## Abstract

### Background

In the United States, already-prevalent power outages are increasing in frequency and duration with climate change. Studies from New York State show that power outages may increase hospitalizations for cardiovascular disease (CVD) and respiratory disease in vulnerable populations such as older adults, but exposure data limitations have constrained nationwide studies of power outages and health. Here, we tested if power outages were associated with emergency CVD and respiratory disease-related hospitalizations among older adults in the United States.

### Methods and findings

We developed a national dataset of power outage exposure and identified county-days with ≥1% of customers exposed to 8+ hour power outages in 2018. We leveraged data on 23 million Medicare Fee-For-Service beneficiaries aged 65+ to estimate daily county-level rates of emergency CVD- and respiratory-related hospitalizations. We applied a case-crossover design with a conditional Poisson model to estimate the lagged association (up to 1 week) between daily county-level power outage exposure and cause-specific hospitalization rates. Models controlled for daily temperature, precipitation, and wind speed.

### Results

Power outages were associated with increased emergency CVD and respiratory hospitalizations. The association between power outage and CVD hospitalizations was

which permits unrestricted use, distribution, and reproduction in any medium, provided the original author and source are credited.

**Data availability statement:** Our analysis code is available at https://github.com/NSAPH-Projects/power_outage_national_cvd_hosp. We used individual Medicare data, which cannot be shared publicly due to privacy laws. Contact CMS at https://www.cms.gov/about-cms/contact/cms. The PowerOutage.us data are not public since they are owned by PowerOutage. us, and are available for purchase from https://poweroutage.us/. Our purchase agreement does not allow us to share them. gridMET data are publicly available for download at https://www.climatologylab.org/gridmet.html.

**Funding:** This work was supported by the following sources: National Institute on Aging R01 AG071024 (JAC), National Institute of Environmental Health Sciences P30 ES007033 (JAC), National Institutes of Health R01 ES034021 (DM), National Institutes of Health R01 AG066793 (DM), National Institute on Aging P20 AG093975 (MAK), National Institute of Environmental Health Sciences P30 ES009089 (MAK) (nih.gov) Canadian Institutes of Health Research Doctoral Foreign Study Award (HM) (https://cihr-irsc.gc.ca/), National Heart, Lung, and Blood Institute F31 HL172608 (VD). The funders had no role in study design, data collection and analysis, decision to publish, or preparation of the manuscript.

**Competing interests:** The authors have declared that no competing interests exist.

**Abbreviations:** CI, confidence interval; CMS, Centers for Medicare and Medicaid Services; CVD, cardiovascular disease; DME, durable medical equipment; ICD-10, International Classification of Diseases, 10th Revision; RECORD, Reporting of Studies Conducted using Observational Routinely-Collected Data; RR, rate ratio.

strongest the day after power outage exposure (rate ratio [RR]=1.02, 95% CI: 1.01, 1.03), while the association between outage and respiratory disease was strongest the day of power outage exposure (RR = 1.03, 95% CI: 1.01, 1.04). We estimated this association using county-level power outage data; future studies could use higher spatial resolution data.

## Conclusions

Power outages may increase the risk of CVD and respiratory hospitalizations among US older adults. Improving electricity reliability could support community health and protect older adults from CVD and respiratory disease exacerbations.

## Author summary

### Why was this study done?

- Power outages threaten health when air conditioners, heaters, communication devices, such as phones and laptops, and medical devices, for example, at-home ventilators and oxygen tanks lose power, exposing people to extreme temperatures, causing medical and mobility device failures, and isolating individuals from help.

- Prior studies have shown power outages might increase risk of emergency department visits and hospitalizations because of heat, cold, medical device failure, and isolation.

- Power outages are becoming more common with climate change, electricity use by artificial intelligence, and electrification, so understanding the health risks of power outages is important.

### What did the researchers do and find?

- We estimated risk of hospitalization for older adults (age 65+) on days with 8+ hour power outages compared to days without power outages.

- We found that power outages lasting 8+ hours were associated with an increased risk of hospitalization for cardiovascular and respiratory disease.

- Assuming power outages are causing these hospitalizations, we estimated that there were 4,246 excess hospitalizations in 2018 among US adults 65+ due to power outages.

### What do these findings mean?

- Power outages may place older adults at risk of hospitalization for cardiovascular and respiratory issues because of heat and cold, medical device failure, and social isolation.

PLOS Medicine

- Investing in electricity reliability could improve community health, as could other interventions like backup batteries for medical devices, cellphones, and laptops, or generators to keep air conditioners and heaters on, especially as electrification continues.

- The data used in this study only described power outages by county, instead of individual people. In the future, knowing which individuals experienced a power outage can help us understand the exact risks of power outage for health.

## Introduction

As the climate warms, the incidence and duration of power outages across the US are increasing [1]. US electrical customers experienced an average of 8 hours without power in 2020—the longest duration on record [2]. Aging electrical grid components, already at risk of failure, were not built to withstand previously rare extreme weather events now common with climate change [3,4]. 40%–60% of major outages are now caused by severe weather events [5]. Additionally, extreme heat and cold events will continue to increase electricity use to maintain indoor temperatures, outstripping supply and causing outages [6,7].

Power outages threaten the health of vulnerable populations such as older adults [3,8]. Outages disable air conditioners and heaters and expose those affected to extreme temperatures [9]. This heat and cold exposure may cause or exacerbate cardiovascular disease (CVD) and respiratory illness. Older adults are more likely to suffer health consequences from heat and cold exposure due to aging-related thermoregulation changes [10–12] and preexisting CVD or respiratory disease [13,14]. More than 70% of older adults live with CVD [15]. Additionally, over 3% of older adults use electricity-dependent medical equipment such as ventilators and oxygen tanks at home to treat conditions like chronic obstructive pulmonary disease [3]. For these individuals, loss of electricity can be directly life-threatening. Finally, prolonged loss of electricity to refrigerators, elevators, wheelchairs, and water supply or communication systems can result in stress, injury, dehydration, or isolation. Older adults' increased reliance on mobility devices, elevators, and increased social isolation [16–18] may put them at higher risk than others for outage-related cardiovascular and respiratory illness.

Prior epidemiologic studies in New York State found elevated cardiovascular and respiratory emergency department visits up to one week after power outage exposure for all adults, as well as increased cardiovascular and respiratory hospitalizations and mortality [19–22]. Associations may be stronger among older adults compared to younger adults [20]. Population-level datasets of power outage exposure beyond New York State have only become available since 2020, limiting prior national-scale studies of power outage and health [3,5,23].

We previously assembled a nationwide dataset of hourly county-level power outage exposure from 2018 to 2020 based on data from PowerOutage.us [5,24]. Here, we used these data together with 2018 older adult Medicare Fee-For-Service hospitalization claims to test if daily county-level power outage exposure was associated with emergency cardiovascular and respiratory hospitalization rates among older adults in the United States, using a case-crossover design. We evaluated the impacts of moderate and large-scale power outages and conducted secondary analyses examining effect measure modification by age, sex, Medicaid eligibility, and electricity-dependent durable medical equipment (DME) use. We hypothesized power outages would cause cardiorespiratory hospitalizations among older adults, and risks would be higher among the oldest and Medicaid-eligible adults, and in counties with higher DME use.

## Methods

### Study population

Our study population included Medicare Fee-For-Service beneficiaries aged 65+ and enrolled for at least one month between January 1, 2018, and December 31, 2018. From the Medicare Beneficiary Summary File, we obtained age, sex, Medicaid-eligibility status, and county of residence for all beneficiaries. Medicare Fee-For-Service beneficiaries comprise

approximately 50% of the US population aged 65+. This population may have fewer chronic health conditions, more education and income, and less healthcare utilization compared to Medicare Advantage beneficiaries [25,26].

We used the Medicare Provider Analysis and Review file to access inpatient claims data on all hospitalizations in our study population in 2018, obtained from the Centers for Medicare and Medicaid Services (CMS). We accessed the date of hospitalization, type of hospitalization (emergency, urgent, or planned), and cause of hospitalization via *International Classification of Diseases*, 10th Revision (*ICD*-10) diagnostic codes.

This study was approved by the WIRB-Copernicus Group (single IRB for multi-site project 20210419), the IRBs at Harvard T.H. Chan School of Public Health, and the Columbia University Mailman School of Public Health. Consent was waived since the data were analyzed anonymously.

### Outcome assessment

Using beneficiaries' county of residence, we tabulated the number of Medicare Fee-for-Service beneficiaries for all US counties. We also tabulated daily, county-level counts of urgent and emergency hospitalizations for cardiovascular or respiratory causes based on the hospitalized beneficiary's county of residence. We identified CVD (I00-I99) and respiratory (J00-J99) hospitalizations based on the first five *ICD*-10 diagnostic codes on the record. We included only urgent and emergency hospitalizations (henceforth referred to as "emergency hospitalizations") since we hypothesized that power outages would increase emergency and urgent hospitalization rates, but not scheduled hospitalizations, due to short-term heat, cold, and electricity-dependent medical device disruption.

### Exposure assessment

To assess power outage exposure for 2018, we used nationwide county-level data purchased from PowerOutage.us (POUS) [24]. These data included the number of customers without power every hour by county. 'Customers' referred to residential consumers, such as households or families, and non-residential consumers, such as businesses. Utilities rarely provide total counts of customers served, and thus, these data are often unreliable from POUS. POUS captures counts of customers served when available, but inconsistencies and gaps mean estimates provided by POUS often differ by orders of magnitude across months and years. Instead of relying on these unstable estimates, we 2013–2018 American Community Survey estimates of the number of households and establishments by county to estimate the proportion of state customers in each county. We then multiplied the proportion of customers in each county by the estimated total number of customers served in each state from the Energy Information Administration [27] to estimate the number of county customers.

Substantial exposure data were missing from the POUS dataset. The POUS dataset was created using web scraping, and some utilities did not have websites, or their websites were offline during part or all of the study period. Previously, we conducted a simulation study where we treated missing data as no power outage exposure, because no exposure was the median value in the dataset [28]. When doing so in simulations, missing data biased the results of a study like ours towards the null. When a small percentage (approximately <15%) of hours within each county (county-hours) were missing exposure data, the bias was minimal [29]. When larger amounts (approximately >50%) were missing, the bias was substantial. To balance generalizability and bias in the present study, we excluded counties with >50% of county-hours missing in the POUS data (*n*=907 counties). On average, the remaining counties were missing data for 7% of county-hours. When we included counties were missing 4 or fewer hours of consecutive exposure data, we carried forward the last observation to impute those hours. Our final analytic dataset included counties in 48 states (all counties in HI and AK were excluded) and 2,161 counties, covering 71.1% of 2018 Fee-For-Service Medicare beneficiaries (*N*=23,622,770).

We were interested in understanding the health impacts of prevalent moderate to large-scale power outages, not only large outages caused by extreme weather events. Therefore, in our primary analysis, we considered a county-day exposed to power outage if ≥1% of county customers were without power for 8+ consecutive hours, a definition that may

have substantial exposure misclassification (i.e., up to 99% of customers may be unexposed). To address this, we conducted a secondary analysis for which we assessed power outage exposure based on higher cut points of customers without power (≥3%, ≥5%), scenarios with less exposure misclassification.

We analyzed 8+ hour power outages because we hypothesized that indoor temperatures would change substantially over this time, exposing older adults to heat and cold. Further, many batteries for electricity-dependent medical equipment last 8 hours. During an 8+ hour power outage, electricity-dependent medical device users could experience adverse health effects without their equipment. We also chose this definition since prior studies have evaluated the health impacts of similar-size outages [19–21,30].

Because there is no literature on the health-relevant duration of power outage beyond epidemiologic studies identifying health impacts following outages of certain lengths [19–21,30–32], we conducted a sensitivity analysis on the power outage duration. We evaluated the effects of 4+ hour outages and 12+ hour outages on CVD and respiratory hospitalization rates. We also conducted a sensitivity analysis using a continuous metric of "daily number of hours without power" (hours where ≥1% of the population was without power) to test for a threshold effect (where only outages longer than a certain duration caused health effects).

## Study design

We used a time-stratified case-crossover design with a conditional Poisson model [33] to analyze the association between daily county-level power outage exposure and CVD or respiratory hospitalization rates. We modeled CVD and respiratory hospitalizations separately because heat, cold, loss of power to medical devices, and dehydration may affect these outcomes differently [34–36]. For each case day (i.e., a county-day with a non-zero hospitalization count), we selected control days from the same county in the month and on the same day of the week. Since our study included only days from 2018, control days were also matched on year. This matching controlled for time-invariant confounders such as county-level socioeconomic factors that might influence both power outage and cardiorespiratory hospitalization rates, as well as long-term, seasonal, and day-of-week trends. We tested for overdispersion in the outcome by running a quasi-Poisson model, but detected none, and thus proceeded with a traditional Poisson model.

## Statistical analysis

We controlled for meteorological confounders such as temperature, precipitation, and wind speed (as a proxy for cyclones, tornadoes, and other storms). These factors influence both power outage and hospitalization rates. We used daily county-level maximal temperature, average wind speed, and total precipitation measures from gridMET, a dataset of daily high-spatial resolution surface meteorological data [37]. We included maximal temperature flexibly in our models as a natural spline with 3 degrees of freedom based on the known non-linear relationship between temperature and hospitalizations, and controlled for lagged effects of temperature up to 1 week after exposure [29]. To determine how flexibly to control for wind speed and precipitation, we examined the relationships between these two meteorologic factors and CVD and respiratory hospitalization rates separately. We ran several test models with splines on precipitation and wind speed with varying degrees of flexibility (linear and 2–4 degrees of freedom) and tested model fit using the quasi-Akaike Information Criterion (qAIC). We controlled for these confounders in our analytic power outage models with the qAIC-determined degree of flexibility. In respiratory hospitalization models, we controlled for same-day precipitation linearly, and in CVD models, with 2 degrees of freedom. We modeled same-day wind speed exposure with 3 degrees of freedom for both outcomes.

We hypothesized that there would be lagged effects of power outage on CVD and respiratory hospitalizations. Power outage exposure had $\rho = 0.2$ autocorrelation. We included distributed lag terms up to 6 days after power outage exposure and constrained these terms [38]. We tested 3–5 degrees of freedom on the lag dimension (1–3 knots), and we compared model fit using qAICs. We found that for CVD outcomes, 5 degrees of freedom across the lag dimension produced the best model fit, and for respiratory hospitalizations, 3 degrees of freedom resulted in the best model fit.

We conducted secondary analyses for power outages affecting ≥3% or ≥5% of county customers, rather than ≥1%.

In an additional secondary analysis, we tested the relationship between continuous daily county-level number of hours without power and hospitalizations rates to test for possible threshold effects. We used constrained non-linear lag terms for power outage exposure in a conditional Poisson model as described above. To test for threshold effects, we compared models with a linear exposure-response function to those with a natural spline exposure-response function with 3 degrees of freedom. We also tested models with 3–6 degrees of freedom on the lag dimension and used qAICs to find the best-fitting model among these eight model options.

Finally, to express our main analysis results in absolute terms, we calculated the population attributable fraction of cardiovascular and respiratory hospitalizations due to power outage and used it to estimate the excess cardiovascular and respiratory hospitalizations associated with power outages in the Medicare Fee-For-Service population in 2018.

## Sensitivity analyses

We ran a sensitivity analysis testing if last observation carried forward versus replacing data missing for fewer than 4 hours with 0 (no exposure, the mean, median, and mode value of exposure in the dataset) affected our results.

Because of the large proportion of missing data in the POUS dataset, we also conducted a sensitivity analysis, limiting our analysis to counties with 20% of data missing or less.

Finally, because of wildfire-related power shutoffs, wildfire smoke could be a confounder. We conducted a sensitivity analysis using estimates of wildfire fine particulate matter ($PM_{2.5}$) from Childs and colleagues 2022 [39] to categorize county-days as exposed or unexposed to wildfire smoke. We considered a county-day exposed if the concentration of wildfire smoke was >0 µg/m$^3$ and adjusted for this binary 'smoke-day' variable in our analysis.

## Testing for effect modification

We tested for effect modification in the association of power outage exposure on CVD and respiratory outcomes by individual age, sex, and poverty status. We stratified analyses by age (65–75 and 75+), sex (male and female), and dual eligibility for Medicaid and Medicare as a proxy for poverty status. We also tested for effect modification by the percentage of total Medicare beneficiaries using DME by county. We calculated the percentage of DME users by county, stratified analyses by quartile, and compared effects between the 1st and 4th quartiles. We estimated DME use with emPOWER data [40], which provide the number of DME users among Medicare beneficiaries (all, including Fee-For-Service, Medicare Advantage, and those <65) and the total number of beneficiaries by county.

Because power outage may affect health by exposing people to extreme indoor temperatures, we also aimed to test if the effect of power outage on CVD and respiratory hospitalizations changed further on anomalously hot and anomalously cold days. Because we are not aware of existing methods for distributed lags for interactions, we modeled the association between a single lag of power outage exposure and hospitalizations for these effect modification analyses. We selected lags that corresponded to the day with the highest effect estimate for each of our two outcomes. We used the same conditional Poisson model and case-crossover as in the main analysis.

For CVD hospitalizations, we modeled the effect of 8+ hour power outage lagged by two days and same-day anomalously hot temperatures. To test for effect modification on hot days, we included a binary term for anomalously hot days and an interaction term between anomalously hot days and power outage. We conducted a separate analysis to assess for effect modification by cold, including a binary term for same-day anomalously cold days and an interaction term between anomalously cold days and lagged power outage. For respiratory hospitalizations, we took the same approach, but used same-day power outage rather than lagged exposure.

To conduct these analyses, first, we identified anomalously hot and cold days using both a threshold and an absolute cutoff. The temperature on hot days had to fall above both the percentile-based threshold and absolute cutoff and below both for cold days. Using data from 1970 to 2010, we identified the 15th and 85th percentile temperatures for each county.

We then categorized days in our study period (2018) as anomalously hot or cold if they were above the 85th or below the 15th percentile temperatures, and hotter than 24 °C or colder than 0 °C (absolute cutoffs), respectively.

We conducted analyses in R 4.4.1, using R packages *gnm* [41], *splines* [42], and *dlnm* [43]. Our analysis code is available at https://github.com/NSAPH-Projects/power_outage_national_cvd_hosp. This study is reported as per the Reporting of Studies Conducted using Observational Routinely-Collected Data (RECORD) guideline (S1 RECORD Checklist).

## Results

In this 2018 county-level case-crossover analysis, we used daily county-level counts of total urgent and emergency CVD and respiratory-related hospitalizations and power outage exposure in a conditional Poisson model to estimate the association between power outage exposure and hospitalization rates up to 1 week later. We considered a county-day exposed to power outage if ≥1% of county customers were without power for 8 or more consecutive hours.

We included 2,161 US counties, covering 71.1% of older adult Medicare Fee-For-Service beneficiaries aged 65+ (*N* = 23,645,101). These counties experienced a median of 2 (Interquartile Range = 4; 5th percentile = 0; 95th percentile = 15) 8+ hour power outages affecting ≥1% of customers in 2018 (Fig 1). The number of Medicare Fee-For-Service beneficiaries per county ranged from 8 to 252,004, and outages affecting ≥1% of county customers on average impacted ≥106 beneficiaries. 1.3% of county-person-days had a power outage (*n* = 23,645,101 county-person-days) (Table 1). On average, more outages occurred on colder and windier days (S1 Table).

The 2018 mean annual CVD hospitalization rate was 1,351 per 10,000 beneficiaries, and the respiratory rate was 805 per 10,000 beneficiaries (Fig 1). The most common causes of emergency CVD hospitalization were primary hypertension (I10), hypertensive heart and chronic kidney disease with heart failure (I30), and hypertensive heart disease with heart failure (I110). The most common causes of emergency respiratory hospitalization were acute respiratory failure with hypoxia (J96.01), acute chronic obstructive pulmonary disease exacerbation (J44.1), and unspecified chronic obstructive pulmonary disease (J44.9).

### CVD hospitalizations

**Main analysis.** In our main analysis, we used a case-crossover design with a conditional Poisson model to test the association between 8+ hour power outage and emergency CVD hospitalization rates up to 1 week after power outage exposure. We found increases in CVD-related hospitalizations 1–3 days after power outage exposure (Fig 2). Outage exposure was not associated with increased hospitalizations on other lag days. One day following power outage exposure, the CVD hospitalization rate was, on average, 2.0% (95% confidence interval (CI): 1.3%, 2.6%) higher compared to days after no exposure (S1 Table).

We also analyzed larger outages affecting ≥3% or ≥5% of county customers. Outages affecting ≥3% of county customers had a stronger association with next-day CVD hospitalization rates than outages affecting ≥1% of county customers (Fig 2). Outages affecting ≥5% of customers were associated with even higher next-day hospitalization rates compared to outages affecting ≥3% or ≥1% of county customers, though the confidence intervals for all of these point estimates overlapped. For outages affecting ≥3% of county customers, the day after outage CVD rates were 2.7% (95% CI: 1.8%, 3.6%) higher than on days after no exposure. For outages affecting ≥5% of the population, rates were 3.1% (95% CI: 2.1%, 4.2%) higher than on days after no exposure.

**Sensitivity analyses.** Because there is no information on the health-relevant duration of power outages, we conducted sensitivity analyses evaluating the impact of 4+ and 12+ hour outages on CVD and respiratory hospitalization rates. For 4+ hour and 12+ hour outages, we observed similar results to 8+ hour outages. Hospitalizations were elevated on lag days 1–3 and 6. The effect estimates for 12+ hour outages on CVD hospitalizations were larger than for 8+ hour outages, and 8+ hour outage estimates were larger than 4+ hour outage effects (S1 Fig).

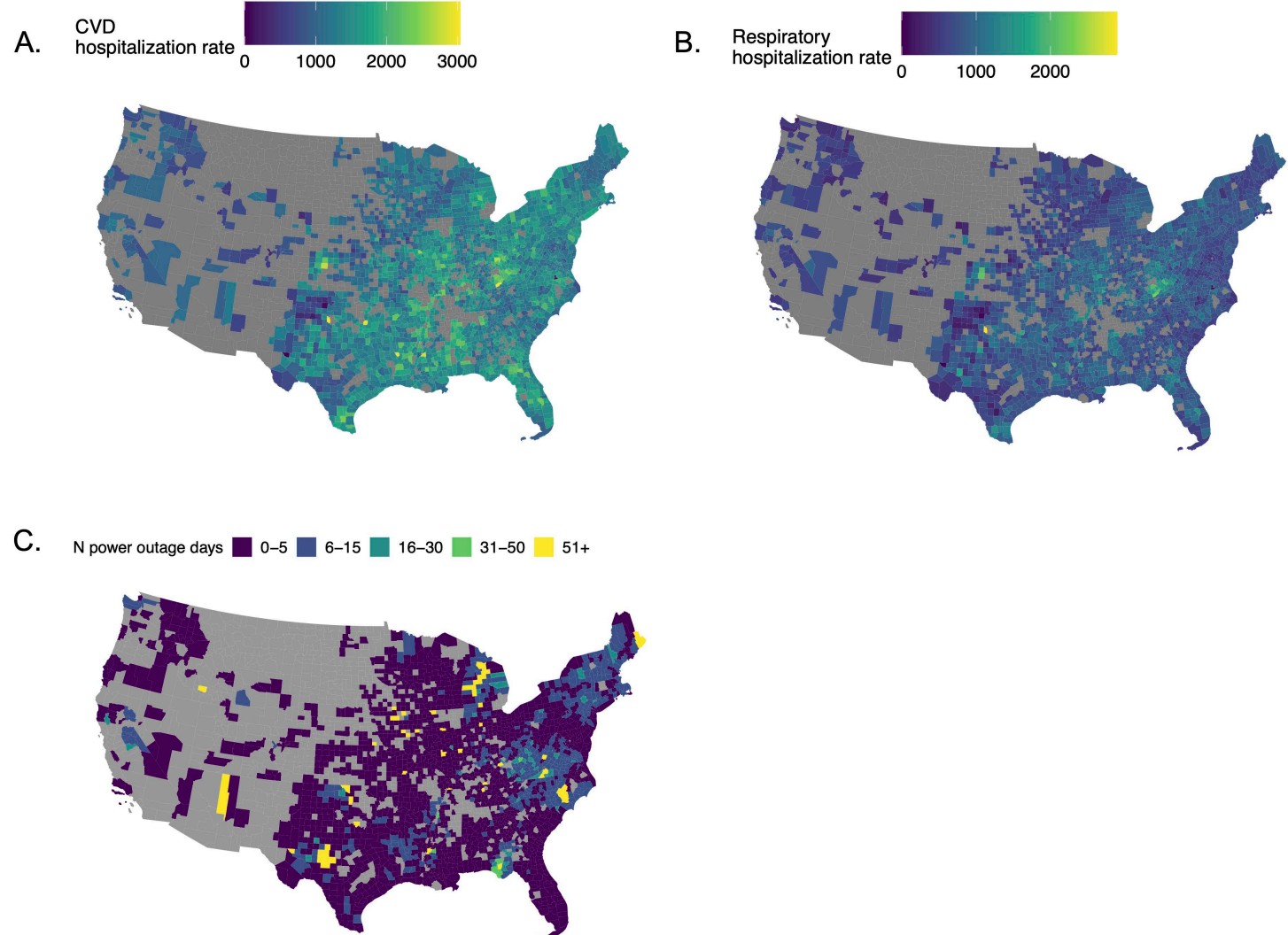

**Fig 1. US county-level hospitalization rates and power outage counts in 2,018 in the 2,161 counties included in main analysis. A**. Medicare Fee-For-Service county-level cardiovascular hospitalization rate per 10,000 beneficiaries. **B**. Medicare Fee-For-Service county-level respiratory hospitalization rate per 10,000 beneficiaries. **C**. Number of power outages in 2018. Basemaps from the U.S. Census Bureau (https://www.census.gov/geographies/mapping-files/time-series/geo/tiger-line-file.2018.html#list-tab-790442341).

We also modeled the relationship between the continuous daily county-level number of hours without power and CVD hospitalization rates to test for a possible threshold effect where outages needed to last a certain duration to cause hospitalizations. We did not identify a threshold; the best-fitting model was linear for the association between number of hours without power and CVD hospitalizations. For every additional hour without power, the next-day CVD hospitalization rate increased by 0.1% and, therefore, by 2.4% for 24 hours without power (S2A Fig).

Last observation carried forward versus replacing data missing for fewer than 4 hours with 0 (no exposure, the mean, median, and mode value of exposure in the dataset) made no difference to our results (S3 Fig).

Restricting to counties with 80% or higher coverage (compared to 50% or higher in our main analyses) did not change our results (S4 Fig).

Our results remained unchanged when we adjusted for wildfire smoke days (S6 Fig).

**Table 1. Distribution of power outages by 2018 Medicare Fee-For-Service study population sociodemographic characteristics.**

| Number of county beneficiaries by category | | Mean percent of county-person-days with 8+ hour outage affecting ≥1% of county customers[a] |
|---|---|---|
| **All** | | |
| | 23,645,101 | 1.3% |
| **Sex** | | |
| Male | 10,824,475 | 1.3% |
| Female | 12,820,626 | 1.2% |
| **Age, years** | | |
| 75 or older | 9,794,477 | 1.3% |
| 65–75 | 13,850,624 | 1.2% |
| **Medicaid eligibility** | | |
| Eligible | 2,620,107 | 0.8% |
| Not eligible | 21,024,994 | 1.7% |
| **County-level Medicare beneficiaries using durable medical equipment** | | |
| Quartile 1 | 14,204,467 | 0.8% |
| Quartile 2 | 5,347,182 | 1.7% |
| Quartile 3 | 2,670,518 | 2.0% |
| Quartile 4 | 1,422,934 | 2.0% |

[a]Mean{total person-days exposed in each county/total person-days in study period in each county} for each category (male/female, 65–75, etc.).

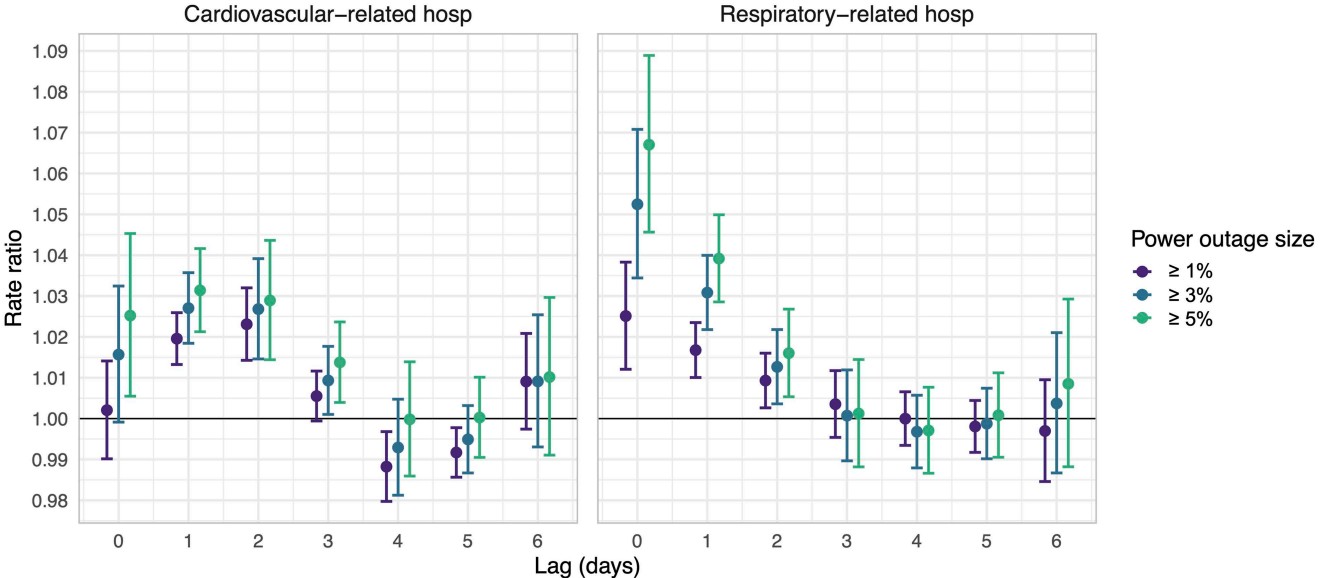

**Fig 2. Rate ratios (circles) and 95% confidence intervals (bars) for the association between county-level 8+ hour power outage exposure and cardiovascular- and respiratory-related hospitalizations in US 2018 Medicare Fee-For-Service beneficiaries for outages affecting ≥1%, ≥3%, and ≥5% of county electrical customers.** Estimates are from conditional Poisson regression models adjusted for daily wind speed, temperature, and precipitation.

### Respiratory hospitalizations

**Main analysis.** In our main analysis testing the association between 8+ hour power outage exposure and emergency respiratory hospitalization rates, we found same-day increases in respiratory-related hospitalizations, as well as increases on lag days 1 and 2 (Fig 2). In contrast to CVD hospitalizations, for respiratory-related hospitalizations, we observed the strongest association on the same day as the power outage rather than the day after. On the same day of power outage exposure, the respiratory hospitalization rate was 2.5% (95% CI: 1.2%, 3.8%) higher than on unexposed days.

Outages affecting ≥3% of county customers resulted in stronger associations with same-day respiratory hospitalization rates than outages affecting ≥1% of county customers (Fig 2). Outages affecting ≥5% of customers were associated with even higher same-day respiratory hospitalization rates compared to outages affecting ≥3% or ≥1% of county customers. For outages affecting ≥3% and ≥5% of county customers, same-day respiratory hospitalization rates were 5.2% (95% CI: 3.4%, 7.1%) and 6.7% (95% CI: 4.6%, 8.9%) higher, respectively, compared to rates on unexposed days.

**Sensitivity analysis.** For sensitivity analyses evaluating the impact of 4+ and 12+ hour outages on respiratory hospitalizations, we found the strongest association for respiratory hospitalizations following 12+ hour outage exposure. Effect size increased from 4+ to 8+ to 12+ hour outage durations. Respiratory hospitalization rates were 1.2% higher (95% CI: 0.2%, 2.2%) the day of 4+ hour power outage exposure, 2.5% higher (95% CI: 1.2%, 3.8%) higher the day of 8+ hour outage exposure, and 3.2 times higher (95% CI: 1.7%, 4.6%) the day of 12+ hour outage exposure compared to unexposed days.

We also modeled the relationship between continuous number of hours without power and respiratory hospitalization rates. Like CVD, the best-fitting model indicated a linear relationship between number of hours without power and respiratory hospitalizations. For every additional hour without power, the next-day respiratory hospitalization rate increased by 0.11% or 2.6% following 24 hours of power outage (S2 Fig).

As with CVD hospitalizations, last observation carried forward versus replacing data missing for fewer than 4 hours with 0 made no difference to our results (S3 Fig), restricting to counties with 80% or higher coverage (compared to 50% or higher in our main analyses) did not change our results (S4 Fig), and adjusting for wildfire smoke days did not change our results (S6 Fig).

### Effect modification

We tested for effect modification of the relationship between power outage and CVD and respiratory hospitalizations by age, sex, dual-Medicaid eligibility, and percentage of county Medicare beneficiaries using DME. We used Medicaid eligibility as a proxy for poverty. Power outage exposure was distributed similarly across potential effect modifiers (Table 1). Overall, we did not observe effect modification by age, sex, or Medicaid eligibility.

For DME use, the association between power outages and respiratory hospitalizations appeared stronger in counties with smaller percentages of DME users (quartile 1 of DME use) compared to counties with larger percentages of DME users (quartile 4 of DME use). Respiratory hospitalizations remained elevated in counties with quartile 1 DME use for two days after power outage, while in counties with fourth quartile DME use, hospitalizations were elevated only on the day of power outage (Fig 3). The effect of power outage on CVD hospitalizations was not modified by county-level DME use.

When we tested if the effect of power outage on CVD hospitalizations differed on anomalously hot days, we observed a larger power outage-associated increase in CVD hospitalizations on hot days compared to non-hot days (S3 Table). On cold days, there was no additional effect of power outage on CVD hospitalizations. We found no additional effect of power outage on respiratory hospitalization on hot days. The effect of power outage on respiratory hospitalizations was attenuated slightly on anomalously cold days (S3 Table).

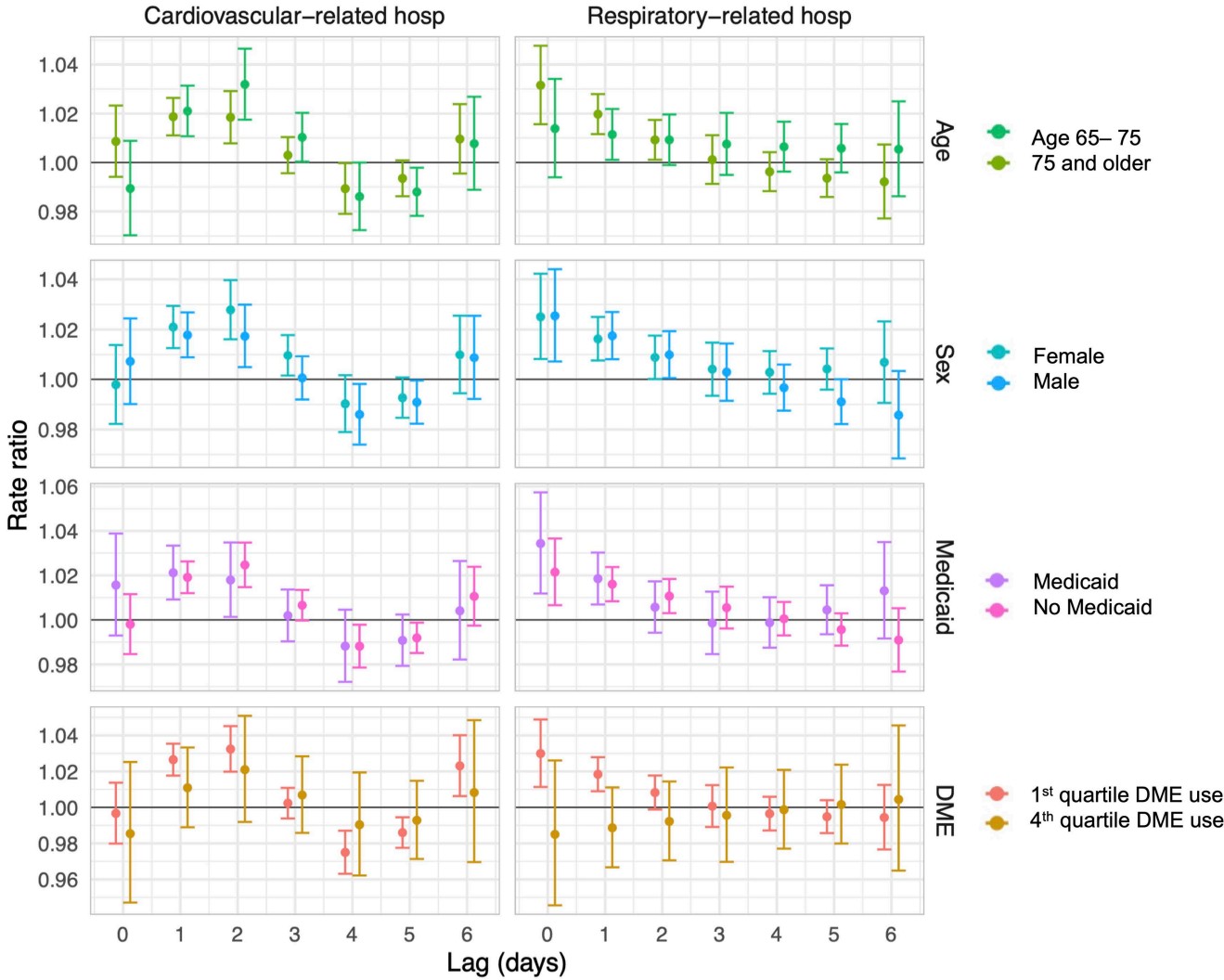

**Fig 3. Rate ratios (circles) and 95% confidence intervals (bars) for the association between county-level 8+ hour power outage exposure and cardiovascular- and respiratory-related hospitalizations in US 2018 Medicare Fee-For-Service beneficiaries for outages affecting ≥1% of county electrical customers, stratified by potential effect modifiers: age, sex, dual-Medicaid eligibility, and county-level durable medical equipment (DME) use quartile (1st quartile=<0.5%, 4th quartile =>0.8%).** Estimates are from conditional Poisson regression models adjusted for daily wind speed, temperature, and precipitation.

### Excess hospitalizations

With an average of seven 8+ hour power outages per year, a 1.04 RR (95% CI: 1.02, 1.06) increase in CVD-related hospitalizations in the seven-day period following an outage would result in approximately 2,408 (95% CI: 1,105, 3,735) CVD-related hospitalizations due to power outage per year in the Fee-for-Service Medicare population, in the absence of other sources of bias. Similarly, across all lags, a 1.05 RR (1.02, 1.08) increase in respiratory-related hospitalizations with power outage would result in approximately 1,838 (95% CI: 819, 2,884) power outage-attributable respiratory hospitalizations per year in this population.

## Discussion

In this 2018 case-crossover study of over 23 million Fee-For-Service Medicare beneficiaries aged 65+, we found that power outages were associated with increased acute emergency CVD and respiratory hospitalizations. This association was strongest for CVD hospitalizations the day following the power outage, while for respiratory hospitalizations, the same-day association was strongest. As expected, larger outages affecting ≥3% or ≥5% versus ≥1% of county customers had stronger associations with hospitalization rates. Furthermore, power outages were prevalent. US counties experienced an average of seven 8+hour outages affecting ≥1% of customers in 2018, and shorter outages were even more common. With outage frequency and duration increasing due to climate change, these outages may pose a growing threat to the cardiovascular and respiratory health of older adults.

Several New York State-based studies have reported associations between power outage exposure and increased CVD and respiratory acute care visits, with larger effects for older versus younger adults [19–21,30]. As in our study, Deng and colleagues found the largest increases in CVD emergency department visits the day after exposure, while respiratory visits increased most on the day of exposure. We observed a protective effect on hospitalizations 4–5 days after a power outage, which may be a harvesting effect or an artifact resulting from the use of a natural spline-based model. For larger outages, there was no protective effect on hospitalizations 4–5 days after exposure, making it unclear whether any true protective effect exists. Do and colleagues estimated the association of power outage with CVD hospitalizations in Medicare beneficiaries 65+ in New York State from 2017 to 2018, which overlaps our study period and population [30]. They found elevated emergency CVD hospitalizations one day after power outage exposure, though confidence intervals contained the null. We estimated similar effects with more precision because of our larger study population.

We hypothesized that power outages may lead to CVD and respiratory hospitalizations in older adults due to increased heat exposure, cold exposure, stress, and loss of electricity to life-sustaining medical devices and mobility aids. Power outages may also cause changes in indoor air quality when dehumidifiers, air purifiers, and ventilation systems lose power. The lagged effects of CVD and respiratory hospitalizations due to power outage likely differ, since the lagged effects of high or low temperature and air pollution appear to differ for CVD and respiratory disease [29,44,45].

Further, many power outages are caused by climate-related severe weather like heat waves, winter storms, hurricanes, wildfires, and floods [3,46,47], which likely amplifies health risks [20,48,49]. While we did not directly assess co-exposure to extreme weather in this study, we controlled for temperature, wind, and precipitation as time-varying confounders in our main analysis. We also conducted a secondary analysis where we tested for modification of the effect of power outage on cardiovascular and respiratory hospitalizations by temperature. We analyzed power outages that occurred on anomalously hot or anomalously cold days and found larger effects of power outage on cardiovascular-related hospitalizations on hot days, attenuated effects of power outage on respiratory-related hospitalizations on cold days, and no changes on other days. Enabled by new national datasets of power outage exposure [23,24], further research could better describe the lagged effects of co-exposure to power outage and extreme temperatures, and could investigate if keeping indoor temperatures stable could prevent some of the adverse health effects of power outages on CVD. Future studies should also examine the joint health effects of outages and severe weather.

In this study, larger outages affecting ≥3% or ≥5% versus ≥1% of county customers were associated with higher hospitalization rates. The reason may be 2-fold. First, the effects of larger outages on hospitalization rates may appear stronger because the exposure is measured more accurately. We expect less exposure misclassification when ≥5% of county customers are without power compared to ≥1% of customers. Although the magnitude of bias is unknown, there is no reason to expect that exposure misclassification would be differential with respect to CVD and respiratory hospitalization rates. Any resulting bias, therefore, would be towards the null [50]. Second, larger outages may also cause more hospitalizations because they are community-wide events [3,49]. During a small power outage, older adults may be able to rely on neighbors or other nearby community resources for social support, electricity, heat, or air conditioning. During a larger

outage, fewer places have power, and fewer people can help. Therefore, more individuals may be exposed to the midstream effects of power outage such as heat and cold during these larger outages, potentially increasing hospitalizations.

Though some prior studies have described the health consequences of power outages, the duration of power outage that impacts health remains unknown. In our main analysis, we assessed exposure to 8+ hour power outages, and we conducted sensitivity analyses evaluating 4+ and 12+ hour outages, finding that longer outage durations were associated with higher hospitalization rates. We also modeled the relationship between the county-level number of hours without power and hospitalization rates, and tested for non-linearity that would indicate a threshold effect, i.e., an inflection point where outages longer than a certain duration started impacting hospitalization risk. We did not find evidence for a threshold. We found that even one-hour outages were associated with increased cardiovascular and respiratory hospitalizations, suggesting that even short outages can have substantial population health impacts, especially since shorter outages are much more prevalent.

Finally, we hypothesized that certain subgroups might have worse responses to power outage exposure and tested for effect modification by sex, age, Medicaid eligibility, and DME use. Contrary to our hypotheses, we did not observe effect modification by age, sex, or Medicaid eligibility.

We did observe effect modification by DME use quartile. Counties with higher prevalence of DME use (4th quartile DME use) had lower hospitalization rates after power outage exposure than counties with lower DME use (1st quartile). We hypothesized the opposite: that counties with a higher proportion of DME versus non-DME users would be more vulnerable to health effects from power outage. Several factors may explain these unexpected findings. First, power outages could cause more mortality among DME versus non-DME users, so mortality could act differentially as a competing risk for hospitalization in the DME user group. We are unable to test this directly as we do not have access to individual-level DME use data. Second, DME users may be more prepared for outages compared to non-DME users, with greater access to generators or backup batteries [51], though reports describing preparedness among vulnerable groups are mixed [52]. Third, we measured county DME use based on how many Medicare beneficiaries used any type of DME, including wheelchairs, beds, oxygen equipment, ventilators, augmentative and alternative communication devices, and more. These users are not equally vulnerable during power outages; certain types of DME use could indicate better access to healthcare, higher adaptive capacity, or a higher likelihood of residence in skilled nursing facilities with backup power. Finally, due to data limitations, the counts used to generate our DME use quartiles are based on the full Medicare population, while only Medicare Fee-For-Service beneficiaries comprise our study population.

Our study had several limitations. We assessed county-level power outage exposure, since no national finer-resolution power outage exposure data are available. We also aggregated ZIP code tabulation area-level counts of hospitalizations to the county level. Our ecological study was subject to the ecological fallacy and potential time-varying confounding bias, meaning that the relationship between power outage and hospitalizations observed at the county level may not hold at the individual level, despite its theoretical plausibility and community concerns about power outage and adverse health outcomes [53–55]. At this time, no individual-level nationwide power outage data are available.

In our main analysis, we considered a county-day exposed to power outage if ≥1% of county customers were without power for 8+ consecutive hours, a definition that may have substantial exposure misclassification (up to 99% of customers may be unexposed). This misclassification likely biased study results towards the null, but the magnitude of bias remains unknown. When we assessed power outage exposure based on higher cut points of customers without power (≥3%, ≥5%), scenarios with less exposure misclassification, the observed effect estimates were stronger. Future studies could collaborate with utilities to obtain finer-resolution power outage data or use satellite imagery to identify exact outage boundaries of long-duration outages lasting into the night [56,57] to improve exposure assessment.

Because POUS did not supply reliable counts of electrical customers by county, we estimated the number of electrical customers using American Community Survey and Energy Information Administration data. We do not know how accurate

these estimates are, and, to our knowledge, no better data exist. If these estimates were not accurate, this may have introduced bias (likely non-differential) into our effect estimates.

Next, power outages may be more common in high social vulnerability communities [5]. High social vulnerability counties may have been overrepresented in the exposed group in our study, and the health effects of power outages may be stronger in these places. This means that our effect estimates may not be generalizable to counties with lower social vulnerability, and we may have overestimated the effect of power outage on hospitalization. At the same time, our study population was comprised of Medicare Fee-For-Service beneficiaries, who may be less vulnerable to health effects from power outages due to higher individual-level socioeconomic status and fewer chronic health conditions compared to Medicare Advantage beneficiaries, which may have led to underestimation of the population-level effect.

Additionally, we assessed DME use as a potential effect modifier at the county level. Because counties are large and diverse, this likely impacted our ability to detect effect modification. This was also US-based study. Electrical grid infrastructure, social cohesion, and healthcare access vary substantially by country, and it is unclear whether our results generalize outside the United States. Other studies are needed to evaluate the health impacts of power outages internationally. Further, we were unable to distinguish between planned and unplanned power outages, which may have different effects on hospitalizations.

Finally, the POUS dataset we used to assess exposure is missing substantial data. We excluded counties missing more than 50% of customer-hours in 2018 to balance generalizability and bias from missing data based on our prior simulation study [28]. Many counties missing >50% of exposure data were rural with low customer counts, and clustered in certain geographic regions, limiting our ability to generalize to these populations. Other studies of power outages and health have found differential effects by urban or rural status, with larger-in-magnitude associations of outages and health in urban areas [30,32].

In this US-wide study of power outage exposure and health, we found that power outages were associated with increased acute CVD and respiratory hospitalization rates among 23 million older adult Medicare beneficiaries. Beneficiaries in our study experienced broad exposure: on average living in counties with seven 8+ hour power outages in 2018, a number likely to increase further with climate change. Heat, winter storms, or other climate-related weather events causing and co-occurring with power outages likely amplify cardiorespiratory health impacts and must be evaluated in future research. Improving electricity reliability represents a key opportunity to support community health and protect older adults from CVD and respiratory disease exacerbations.

## Supporting information

**S1 Table. Distribution of power outage exposure by potential confounders for main analysis of county-level 8+ hour power outage exposure and CVD and respiratory hospitalizations in US 2018 Fee-For-Service Medicare beneficiaries.**
(DOCX)

**S2 Table. S2A Table:** Rate ratios and 95% confidence intervals for the association between county-level 8+ hour power outage exposure and CVD and respiratory hospitalizations in US 2018 Fee-For-Service Medicare beneficiaries for outages affecting ≥1%, ≥3%, and ≥5% of county electrical customers. Estimates are from conditional Poisson regression models adjusted for daily wind speed, temperature, and precipitation. CVD, cardiovascular disease; Resp, respiratory disease. **S2B Table:** Rate ratios and 95% confidence intervals for effect modification analysis of the association between county-level 8+ hour power outage exposure and CVD and respiratory hospitalizations in US 2018 Fee-For-Service Medicare beneficiaries for outages affecting ≥1% of county electrical customers. Estimates are from conditional Poisson regression models adjusted for daily wind speed, temperature, and precipitation and stratified by effect modification categories. CVD, cardiovascular disease; Resp, respiratory disease.
(DOCX)

**S3 Table. Rate ratios and 95% confidence intervals for the association between county-level 8+ hour power outage exposure and cardiovascular- and respiratory-related hospitalizations in US 2018 Medicare Fee-For-Service beneficiaries on anomalously hot and anomalously cold days.** Estimates are from conditional Poisson regression models adjusted for daily wind speed, temperature, and precipitation.
(DOCX)

**S1 Fig. Rate ratios (circles) and 95% confidence intervals (bars) for the association between county-level 8+ hour power outage exposure and CVD and respiratory hospitalizations in US 2018 Fee-For-Service Medicare beneficiaries for 4+, 8+, and 12+ hour power outages affecting ≥1% of county customers.** Estimates are from conditional Poisson regression models adjusted for daily wind speed, temperature, and precipitation.
(DOCX)

**S2 Fig. Rate ratios (blue line) and 95% confidence intervals (shading) for the association between county-level daily number of hours without power and cardiovascular- and respiratory-related hospitalizations in US 2018 Medicare Fee-For-Service beneficiaries.** Estimates are from conditional Poisson regression models adjusted for daily wind speed, temperature, and precipitation. Results are from the best fitting model tested based on qAIC comparison, with a linear relationship between number of hours without power and respiratory hospitalizations or CVD hospitalizations, and 4 degrees of freedom on the lag dimension.
(DOCX)

**S3 Fig. Rate ratios (circles) and 95% confidence intervals (bars) for the association between county-level 8+ hour power outage exposure and cardiovascular- and respiratory-related hospitalizations in US 2018 Medicare Fee-For-Service beneficiaries for outages affecting ≥1%, ≥3%, and ≥5% of county electrical customers.** In this sensitivity analysis, instead of last observation carried forward, some missing values were replaced with 0. Estimates are from conditional Poisson regression models adjusted for daily wind speed, temperature, and precipitation.
(DOCX)

**S4 Fig. Rate ratios (circles) and 95% confidence intervals (bars) for the association between county-level 8+ hour power outage exposure and cardiovascular- and respiratory-related hospitalizations in US 2018 Medicare Fee-For-Service beneficiaries for outages affecting ≥1%, ≥3%, and ≥5% of county electrical customers, in counties with 20% of data missing or less.** Estimates are from conditional Poisson regression models adjusted for daily wind speed, temperature, and precipitation.
(DOCX)

**S5 Fig. Directed acyclic graph describing hypothesized causal relationships between power outage, hospitalizations among older adults, and potential confounders.**
(DOCX)

**S6 Fig. Rate ratios (circles) and 95% confidence intervals (bars) for the association between county-level 8+ hour power outage exposure and cardiovascular- and respiratory-related hospitalizations in US 2018 Medicare Fee-For-Service beneficiaries for outages affecting ≥1%, ≥3%, and ≥5% of county electrical customers.** Estimates are from conditional Poisson regression models adjusted for daily wind speed, temperature, precipitation and wildfire $PM_{2.5}$.
(DOCX)

**S1 RECORD Checklist. RECORD checklist required by PLOS Medicine.** Checklist of items, extended from the STROBE statement, that should be reported in observational studies using routinely collected health data.
(DOCX)

## Author contributions

**Conceptualization:** Marianthi-Anna Kioumourtzoglou, Joan A. Casey.

**Data curation:** Heather McBrien, Daniel Mork, Vivian Do, Marianthi-Anna Kioumourtzoglou, Joan A. Casey.

**Formal analysis:** Heather McBrien, Vivian Do, Joan A. Casey.

**Funding acquisition:** Marianthi-Anna Kioumourtzoglou, Joan A. Casey.

**Investigation:** Heather McBrien, Marianthi-Anna Kioumourtzoglou, Joan A. Casey.

**Methodology:** Heather McBrien, Vivian Do.

**Project administration:** Heather McBrien.

**Software:** Heather McBrien.

**Supervision:** Daniel Mork, Marianthi-Anna Kioumourtzoglou, Joan A. Casey.

**Visualization:** Heather McBrien.

**Writing – original draft:** Heather McBrien, Daniel Mork, Vivian Do, Marianthi-Anna Kioumourtzoglou, Joan A. Casey.

**Writing – review & editing:** Heather McBrien, Daniel Mork, Vivian Do, Marianthi-Anna Kioumourtzoglou, Joan A. Casey.

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
