## [Editor Report · Decision Letter 0]

29 Apr 2025

Dear Dr McBrien,

Thank you for submitting your manuscript entitled "Power outages and cardiovascular and respiratory hospitalizations among US older adults" for consideration by PLOS Medicine.

Your manuscript has now been evaluated by the PLOS Medicine editorial staff and I am writing to let you know that we would like to send your submission out for external assessment.

However, we first need you to complete your submission by providing the metadata that are required for full assessment. To this end, please login to Editorial Manager where you will find the paper in the 'Submissions Needing Revisions' folder on your homepage. Please click 'Revise Submission' from the Action Links and complete all additional questions in the submission questionnaire.

Please re-submit your manuscript within two working days, i.e. by May 01 2025 11:59PM.

Once your full submission is complete and has passed all checks it will be sent out for external assessment.

Kind regards,

Richard Turner, PhD

Consulting Editor, PLOS Medicine

plosmedicine@plos.org

---

## [Decision Letter · Decision Letter 1]

9 Jul 2025

Dear Dr McBrien,

Sincere apologies for the delay in getting back to you with a decision which was due to challenges in securing the necessary reviewers. Many thanks for submitting your manuscript "Power outages and cardiovascular and respiratory hospitalizations among US older adults" (PMEDICINE-D-25-01514R1) to PLOS Medicine. The paper has been reviewed by subject experts and a statistician; their comments are included below and can also be accessed here: [LINK]

As you will see, the reviewers find your work of great interest but they would like clarification of several methodological points, for example with regards to missing data, and they have also raised concerns about potential sources of bias which we would like to see resolved in full. After discussing the paper with the editorial team and an academic editor with relevant expertise, I'm pleased to invite you to revise the paper in response to the reviewers' comments. We plan to send the revised paper to some or all of the original reviewers, and we cannot provide any guarantees at this stage regarding publication.

We ask that you submit your revision by Jul 30 2025 11:59PM. However, if this deadline is not feasible, please contact me by email, and we can discuss a suitable alternative.

Don't hesitate to contact me directly with any questions (acunha@plos.org).

Best regards,

Andreia

Andreia Cunha, PhD

Senior Editor

PLOS Medicine

acunha@plos.org

Comments from the academic editor:

1. Agree with R2 about improving the description of the total number of outages (7 outages, SD=29, of 8+ hours). The huge right skew renders mean/SD uninterpretable, so median/IQR is needed.

2. Also, even if removing the mean/SD, as R2 points out I just think the phrasing of this statistic is not easy for readers: "counties experienced an average of 7 (standard deviation, SD=29) 8+ hour power outages affecting ≥1% of customers in 2018." I have lived in rural U.S., and I had maybe 1 of these outages per year. So it does not resonate with me. The issue is definitional -- the reader's mind automatically interprets this to mean that each individual in the U.S> experiences 8 outages per year. But the 1% bit of the statistic is doing a lot of work here -- it means that only a tiny number of households, different each time, are affected. The more useful statistic is the median + IQR of power outages experienced by a given person/household per year.

3. Would a DAG be helpful in illuminating the assumptions in this causal inference paper? In particular, being transparent about which confounders are and are not controlled for?

4. Could the authors describe their results in absolute terms? The Poisson output of IRR is useful as a relative measure. But ultimately, it is hard to get a sense for the absolute impact as all of the IRRs are tiny even if statistically significant.

Comments from the reviewers:

Reviewer #1: This is a study on a novel research question. Generally the study appears to be well conducted and reported. Particularly, Figure 2 is very clear and illustrated the results beautifully. My methodological comments are:

1. It appears the study used ecological case-cross over design, where the 'ecological' part is that the unit of analysis is based on county rather than individuals. If my interpretation is correct please be explicit about the study's ecological nature, and discuss the limitations arises from this, notably ecological fallacy and potentially individual-level time varying confounding

2. More details are required on the selection of 'control' days. It was reported as 'matching on county, day of week, and month'. In that case the control days would be at least 1 year apart from the 'case' days. How far apart are the case and control days are? Would it not be an issue that with year difference, demographic shift would play a non-negligible role?

3. Consider splitting some text in Statistical Analysis into a separate section as 'Study design' for clarity.

4. Consider adding a negative control outcome (e.g. infectious disease?) - this could add confidence to the findings given the ecological nature.

Reviewer #2: See attachment

Michael Dewey

Reviewer #3: This study investigates the association between power outages and emergency hospitalizations for cardiovascular and respiratory diseases among older adults in the US. The authors assembled the first nationwide dataset of power outage exposure and linked it with Medicare claims data from 23 million beneficiaries aged 65 and older at county level. Using a case-crossover design with a conditional Poisson model, they analyzed the lagged effects (up to 1 week) of outages on hospitalization rates. Overall, the study addresses a critical gap in understanding the health impacts of power outages in the US, which are becoming more frequent due to climate change. However, I have several concerns regarding exposure assessment, confounding, and result interpretation.

(1) This study does not have exposure data at a fine spatial resolution such as census tract or ZIP code level. Power outage exposure is assessed at the county level, meaning many individuals within a county may not have actually experienced a power outage. The definition of exposure (≥1% of customers without power for 8+ hours) may have led to substantial exposure misclassification.

(2) The PowerOutage.us (POUS) dataset is incomplete, with many counties missing up to 50% of data. The authors used a simple method to impute county-days with missing 4 or fewer hours, but it is unclear how the remaining missing values were handled. Further discussion is needed on how missing data may have biased the results.

(3) The study uses different sources for the number of affected customers and the total number of customers. This inconsistency may introduce bias in estimating the proportion of the population affected by power outages. The authors should justify this approach or explore alternative sources that ensure consistency.

(4) If vulnerable individuals (e.g., people of lower SES) are overrepresented among the exposed group, while healthier individuals are not affected, the population-level effect may be overestimated. The authors should clarify how they accounted for this potential bias.

(5) Based on Figure 1, most excluded counties are in the Midwest and Western regions, raising concerns about regional representativeness. Please consider focusing on regions with more complete data (e.g., South and Northeast) and assessing whether included and excluded counties are comparable in main characteristics.

(6) The Poisson regression model was adjusted for temperature, precipitation, and wind speed, but it did not account for air pollution and other extreme weather events, such as wildfires and ice storms, which may influence both power outages and hospitalizations. I suggest adjusting for air pollution and a more comprehensive set of meteorological covariates including extreme weather events in sensitivity analyses.

(7) This study did not distinguish between planned and unplanned power outages. Planned outages may have less severe health effects as residents may be better prepared. If possible, the authors should analyze them separately.

(8) The analyses were conducted at the county level given the exposure assessment method. It would be more appropriate to explore effect modification by county-level characteristics such as urban/rural, socioeconomic status, and geographic region. Since the temperature and season data are readily available, please consider implementing the effect modification by temperature and season in the current study.

(9) Since the approach to handling missing data is critical to the results, a sensitivity analysis should be conducted to assess the robustness of the findings when removing data with varying proportions of missingness.

Minor comments:

(1) Please consider including the results for qAIC in supplementary materials for completeness.

(2) In the main analysis results for CVD hospitalizations, the authors report increases in CVD-related hospitalizations 1-3 and 6 days after power outage exposure. However, Figure 2 suggests that significant increases are only observed on lag days 1 and 2. Additionally, Figure 2 indicates that power outages with a size ≥1% appear to have protective effects on CVD-related hospitalizations on lag days 4 and 5 (a similar pattern is observed in Supplemental Figure 1). Could the authors provide an explanation for these findings?

(3) The discussion could include an explanation for the differing lag effects observed for CVD and respiratory-related hospitalizations.

---

* Please upload any figures associated with your paper as individual TIF or EPS files with 300dpi resolution at resubmission; please read our figure guidelines for more information on our requirements: http://journals.plos.org/plosmedicine/s/figures. While revising your submission, please upload your figure files to the PACE digital diagnostic tool, https://pacev2.apexcovantage.com/. PACE helps ensure that figures meet PLOS requirements. To use PACE, you must first register as a user. Then, login and navigate to the UPLOAD tab, where you will find detailed instructions on how to use the tool. If you encounter any issues or have any questions when using PACE, please email us at PLOSMedicine@plos.org.

* Please revise your financial disclosure statement and competing interests information according to the guidance provided in the submission system.

* The Data Availability Statement (DAS) requires revision. For each data source used in your study:

If the data are not freely available, please describe briefly the ethical, legal, or contractual restriction that prevents you from sharing it. Please also include an appropriate contact (web or email address) for inquiries (again, this cannot be a study author).

* The Ethics statement requires revision: please add the approval number and state whether informed consent was obtained or if this was not required or waived please add this information to the methods section stating the reason why.

FIGURES AND TABLES

SUPPLEMENTARY MATERIAL

REFERENCES

OBSERVATIONAL STUDIES

* Abstract: Please include the study design, population and setting, number of participants, years during which the study took place (enrollment and follow up), length of follow up, and main outcome measures.

* Please ensure that the study is reported according to the STROBE or RECORD guidelines - whichever is more appropriate.

* For STROBE (or appropriate STOBE extension) please follow the guidelines (available from: https://www.equator-network.org/reporting-guidelines/strobe) and include the completed STROBE (or STROBE extension) checklist as Supporting Information. Please add the following statement, or similar, to the Methods: "This study is reported as per the Strengthening the Reporting of Observational Studies in Epidemiology (STROBE) guideline (S1 Checklist)." When completing the checklist, please use section and paragraph numbers, rather than page numbers.

* Dor RECORD (available from https://www.record-statement.org) please follow the guidelines and include the completed checklist as Supporting Information. Please add the following statement, or similar, to the Methods: "This study is reported as per the Reporting of Studies Conducted using Observational Routinely-Collected Data (RECORD) guideline (S1 Checklist)." When completing the checklist, please use section and paragraph numbers, rather than page numbers.

* For all observational studies, in the manuscript text, please indicate: (1) the specific hypotheses you intended to test, (2) the analytical methods by which you planned to test them, (3) the analyses you actually performed, and (4) when reported analyses differ from those that were planned, transparent explanations for differences that affect the reliability of the study's results. If a reported analysis was performed based on an interesting but unanticipated pattern in the data, please be clear that the analysis was data driven.

* Please state in the Methods section whether the study had a prospective protocol or analysis plan. If a prospective analysis plan (from your funding proposal, IRB or other ethics committee submission, study protocol, or other planning document written before analyzing the data) was used in designing the study, please include the relevant document(s) with your revised manuscript as a Supporting Information file to be published alongside your study and cite it in the Methods section. A legend for this file should be included at the end of your manuscript. If no such document exists, please make sure that the Methods section transparently describes when analyses were planned, and when/why any data-driven changes to analyses took place. Changes in the analysis, including those made in response to peer review comments, should be identified as such in the Methods section of the paper, with rationale.

---

## [Decision Letter · Decision Letter 2]

17 Oct 2025

Dear Dr. McBrien,

Thank you very much for re-submitting your manuscript "Power outages and cardiovascular and respiratory hospitalizations among US older adults" (PMEDICINE-D-25-01514R2) for review by PLOS Medicine. I am writing on behalf of my colleague Andreia Cunha who is presently away from the office.

I have discussed the paper with my colleagues and the academic editor and it was also seen again by 3 reviewers. I am pleased to say that provided the remaining editorial and production issues are dealt with we are planning to accept the paper for publication in the journal.

[LINK]

We look forward to receiving the revised manuscript by Oct 24 2025 11:59PM.

Sincerely,

Alison Farrell, Ph.D.

Senior Editor

PLOS Medicine

plosmedicine.org

Requests from Editors:

* At this stage, we ask that you include a short, non-technical Author Summary of your research to make findings accessible to a wide audience that includes both scientists and non-scientists. The Author Summary should immediately follow the Abstract in your revised manuscript. This text is subject to editorial change and should be distinct from the scientific abstract. Ideally each sub-heading should contain 2-3 single sentence, concise bullet points containing the most salient points from your study. In the final bullet point of ‘What Do These Findings Mean?’ Please include the main limitations of the study in non-technical language.

Please see our author guidelines for more information: https://journals.plos.org/plosmedicine/s/revising-your-manuscript#loc-author-summary.

* Please confirm that your title complies with to PLOS Medicine's style. Your title must be nondeclarative and not a question. It should begin with main concept if possible. "Effect of" should be used only if causality can be inferred, i.e., for an RCT. Please place the study design ("A randomized controlled trial," "A retrospective study," "A modelling study," etc.) in the subtitle (ie, after a colon).

* Please revise your abstract to comply with our requirements, including format (three sections: Background, Methods and Findings, and Conclusions) and providing all the information relevant to this study type https://journals.plos.org/plosmedicine/s/submission-guidelines#loc-abstract

* Please ensure that the Introduction ends with a clear description of the study question or hypothesis.

* Please ensure that all abbreviations are defined at first use throughout the text.

* Please confirm that all numbers presented in the abstract are present and identical to numbers presented in the main manuscript text.

* Please review your text for claims of novelty or primacy (e.g. 'for the first time') and remove this language. In addition, please check that any use of statistical terms (such as trend or significant) are supported by the data, and if not please remove them.

* Please remove the 'conclusions' subheading from the discussion. Please also remove any other subheadings from the discussion.

* In the abstract, please include the important dependent variables that are adjusted for in the analyses.

* The Data Availability Statement (DAS) requires revision. For each data source used in your study:

* Please indicate how you obtained Medicare information if not available to other researchers.

* Please explain whether you purchased the power outage data.

* Please include the statement on code availability and url in the data availability statement.

* Please provide the name(s) of the institutional review board(s) that provided ethical approval as well as the application identifier.

* Please confirm that the appropriate usage rights apply to the use of this map. Please see our guidelines for map images: https://journals.plos.org/plosmedicine/s/figures#loc-maps . Also note that figures cannot be reproduced from other sources that are not CC-BY.

* Please provide the unadjusted comparisons as well as the adjusted comparisons in all relevant Tables

* Please specify the variables controlled for in all relevant Tables.

*Please revise your references as per PLOS Medicine style.

* The funding statement should include: specific grant numbers, initials of authors who received each award, URLs to sponsors’ websites. Also, please state whether any sponsors or funders (other than the named authors) played any role in study design, data collection and analysis, the decision to publish, or preparation of the manuscript. If they had no role in the research, include this sentence: “The funders had no role in study design, data collection and analysis, decision to publish, or preparation of the manuscript.”

For Observational studies:

* Please ensure that the study is reported according to the STROBE guideline, and include the completed STROBE checklist as Supporting Information. Please add the following statement, or similar, to the Methods: "This study is reported as per the Strengthening the Reporting of Observational Studies in Epidemiology (STROBE) guideline (S1 Checklist)."

* Did your study have a prospective protocol or analysis plan? Please state this (either way) early in the Methods section.

* Your study is observational and therefore causality cannot be inferred. Please remove language that implies causality and refer to associations instead. Please revise the Abstract accordingly.

* For all observational studies, in the manuscript text, please indicate: (1) the specific hypotheses you intended to test, (2) the analytical methods by which you planned to test them, (3) the analyses you actually performed, and (4) when reported analyses differ from those that were planned, transparent explanations for differences that affect the reliability of the study's results. If a reported analysis was performed based on an interesting but unanticipated pattern in the data, please be clear that the analysis was data-driven.

Please refer to the following guidance for Population Health/Registry studies:

Please ensure that the study is reported according to the RECORD guideline (available from https://www.record-statement.org) and include the completed checklist as Supporting Information. Please add the following statement, or similar, to the Methods: "This study is reported as per the Reporting of Studies Conducted using Observational Routinely-Collected Data (RECORD) guideline (S1 Checklist)." When completing the checklist, please use section and paragraph numbers, rather than page numbers.

Please refer to the following guidance for Population-level Health Estimates:

* Please report your data according to GATHER and enclose a completed GATHER checklist as a supplementary document. See http://gather-statement.org/ In the checklist please include sufficient text excerpted from the manuscript to explain how you accomplished all applicable items.

For modeling studies, we request that the following guidance be followed (derived from Geoffrey P Garnett, Simon Cousens, Timothy B Hallett, Richard Steketee, Neff Walker. Mathematical models in the evaluation of health programmes. (2011) Lancet DOI:10.1016/S0140-6736(10)61505-X.):

* Please provide a diagram that shows the model structure, including how the disease natural history is represented, the process and determinants of disease acquisition, and how the putative intervention could affect the system.

* Please provide a complete list of model parameters, including clear and precise descriptions of the meaning of each parameter, together with the values or ranges for each, with justification or the primary source cited, and important caveats about the use of these values noted.

* Please provide a clear statement about how the model was fitted to the data including where relevant goodness-of-fit measure, the numerical algorithm used, which parameter varied, constraints imposed on parameter values, and starting conditions.

* For uncertainty analyses, please state the sources of uncertainties quantified and not quantified this can include parameter, data, and model structure.

* Please provide sensitivity analyses to identify which parameter values are most important in the model. Uncertainty estimates seek to derive a range of credible results on the basis of an exploration of the range of reasonable parameter values. The choice of method should be presented and justified.

* Please discuss the scientific rationale for this choice of model structure and identify points where this choice could influence conclusions drawn. Please also describe the strength of the scientific basis underlying the key model assumptions.

Comments from Reviewers:

Reviewer #1: Thank you for addressing my comments. The paper reads very clear now. I have no further comments. Congratulations on completing this study.

Reviewer #2: The authors have addressed my points particularly why they expect the two conditions to be differentially affected.

Michael Dewey

Reviewer #3: Many thanks for the authors' thorough responses, which have fully addressed my previous comments. I particularly like their previous simulation analysis to justify the missing data threshold, the new sensitivity analyses (both on the missing data threshold and the additional control for wildfire smoke), and the effect modification by temperature. Kudos to the authors team on such a nice job in revision. I recommend this paper to be accepted.

[LINK]

---

## [Editor Report · Decision Letter 3]

10 Nov 2025

Dear Dr. McBrien,

Thank you very much for re-submitting your manuscript "Power outages and cardiovascular and respiratory hospitalizations among US Medicare beneficiaries in 2018: a case-crossover study" (PMEDICINE-D-25-01514R3) for review by PLOS Medicine.

I have discussed the paper with my colleagues and I am pleased to say that provided the remaining editorial and production issues are dealt with we are planning to accept the paper for publication in the journal.

The remaining issues that need to be addressed are listed at the end of this email. Please take these into account before resubmitting your manuscript:

[LINK]

In revising the manuscript for further consideration here, please ensure you address the specific points made by the editors. In your rebuttal letter you should indicate your response to the editors' comments and the changes you have made in the manuscript. Please submit a clean version of the paper as the main article file. A version with changes marked must also be uploaded as a marked up manuscript file.

Please also check the guidelines for revised papers at http://journals.plos.org/plosmedicine/s/revising-your-manuscript for any that apply to your paper.

We look forward to receiving the revised manuscript by Nov 17 2025 11:59PM.

Sincerely,

Andreia Cunha, Ph.D.

Senior Editor

PLOS Medicine

plosmedicine.org

Requests from Editors:

* Author Summary: please add a final sentence to your summary stating briefly the study limitations.

* Title: we would suggest starting the title as follows ‘The association between power outages…

* Abstract: please revise the heading Conclusion to Conclusions.

*Please remove significance statement from the manuscript.

* Please make the hypotheses tested clearer at the end of the introduction section, i.e. we tested if power outages were associated with emergency CVD and respiratory disease-related hospitalizations among older adults in the United States.

* Abbreviations – Please define on first use WCG, IQR, COPD, ZCTA, SES, ICD, PM, CI.

* Please check that any use of statistical terms (such as trend or significant) are supported by the data, and if not please remove them.

* Please revise your Data Availability Statement (DAS) and ensure it includes for each data set used in your study:

a) If the data are owned by a third party but freely available upon request, please note this and state the owner of the data set and contact information for data requests (web or email address). Note that a study author cannot be the contact person for the data.

b) If the data are not freely available, please describe briefly the ethical, legal, or contractual restriction that prevents you from sharing it. Please also include an appropriate contact (web or email address) for inquiries (again, this cannot be a study author).

* Please also include the statement on code availability and url in the data availability statement.

*Please add IRB approval numbers to the ethics statement and indicate the form of consent obtained (written/verbal) or the reason that consent was not obtained (e.g. the data were analyzed anonymously)

*Please ensure that the study is reported according to the RECORD guideline (available from https://www.record-statement.org) and include the completed checklist as Supporting Information. Please add the following statement, or similar, to the Methods: "This study is reported as per the Reporting of Studies Conducted using Observational Routinely-Collected Data (RECORD) guideline (S1 Checklist)." When completing the checklist, please use section and paragraph numbers, rather than page numbers. (edited)

[LINK]

---

## [Editor Report · Decision Letter 4]

28 Nov 2025

Dear Dr. McBrien,

Thank you very much for re-submitting your manuscript "The association between power outages and cardiovascular and respiratory hospitalizations among US Medicare beneficiaries in 2018: a case-crossover study" (PMEDICINE-D-25-01514R4) for review by PLOS Medicine.

There are some remaining issues that need to be addressed and are listed below.

1) Please remove the guidance for authors following each question in the Author summary.

2) Please replace the word significantly in the following sentence: Electrical grid infrastructure, social cohesion, and healthcare access vary significantly by country, and it is unclear whether our results generalize outside the United States.

3) Please indicate in the ethics statement if consent was not necessary or if it was waived, and please state the reason (e.g. the data were analyzed anonymously).

4) Please revise your Data Availability Statement (DAS) and ensure it includes for Medicare data, PowerOutage.us data and gridMET data the following:

a) If the data are owned by a third party but freely available upon request, please note this and state the owner of the data set and contact information for data requests (web or email address). Note that a study author cannot be the contact person for the data.

b) if the data are not freely available, please describe briefly the ethical, legal, or contractual restriction that prevents you from sharing it. Please also include an appropriate contact (web or email address) for inquiries (again, this cannot be a study author).

For the Medicare data please follow the guidance in point b.

For the PowerOutage.us data please follow the guidance in point b.

For the gridMET data please follow the guidance in point a.

Please provide a point-by-point response to these points with your revised manuscript.

We look forward to receiving the revised manuscript by Dec 05 2025 11:59PM.

Sincerely,

Andreia Cunha, PhD

Senior Editor

PLOS Medicine

plosmedicine.org

[LINK]

---

## [Editor Report · Decision Letter 5]

22 Jan 2026

Dear Dr McBrien,

On behalf of my colleagues and the Academic Editor, Dr David Flood, I am pleased to inform you that we have agreed to publish your manuscript "The association between power outages and cardiovascular and respiratory hospitalizations among US Medicare beneficiaries in 2018: a case-crossover study" (PMEDICINE-D-25-01514R5) in PLOS Medicine.

Before your manuscript can be formally accepted you will need to complete some formatting changes, which you will receive in a follow up email. I have also made some copy-edits to the Author summary for and Figure legends for consistency - please ensure you review these sections. Please be aware that it may take several days for you to receive this email; during this time no action is required by you. Once you have received these formatting requests, please note that your manuscript will not be scheduled for publication until you have made the required changes.

PRESS

Sincerely,

Andreia Cunha, PhD

Senior Editor

PLOS Medicine